# Characteristics of the Most Cited, Most Downloaded, and Most Mentioned Articles in General Medical Journals: A Comparative Bibliometric Analysis

**DOI:** 10.3390/healthcare8040492

**Published:** 2020-11-18

**Authors:** Ji Hyun Hong, Dae Young Yoon, Kyoung Ja Lim, Ji Yoon Moon, Sora Baek, Young Lan Seo, Eun Joo Yun

**Affiliations:** Department of Radiology, Kangdong Seong-Sim Hospital, Hallym University College of Medicine, Seoul 134-701, Korea; whitehong2@gmail.com (J.H.H.); cosmos95@kdh.or.kr (K.J.L.); thanks1029@hanmail.net (J.Y.M.); sorapig@hanmail.net (S.B.); ylseomd@naver.com (Y.L.S.); yeunjookr@naver.com (E.J.Y.)

**Keywords:** bibliometric analysis, citation, download, altmetrics, general medical journals

## Abstract

We compared the characteristics of the most cited, most downloaded, and most mentioned (the highest Altmetric Attention Score) articles published in general medical journals. We identified the 640 most frequently cited, 662 most frequently downloaded, and 652 most mentioned articles from 48 general medical journals. A comparison was made of the following characteristics of articles in the most cited, most downloaded, and most mentioned articles: medical specialty, publication type, country of origin, year of publication, and accessibility. There was only a 2.5% overlap in these three groups. Original articles were the more frequent among the most mentioned articles, whereas reviews, case reports, and guidelines/consensus statements were more frequent among the most downloaded articles. The most cited articles were more frequently published in 2010 and before, whereas the most downloaded articles were published in 2017−2018. The most mentioned articles were more frequently open-access articles, compared to the most downloaded articles. The most cited were more frequently older, the most downloaded were more frequently recent and educational, and the most mentioned were more frequently original and open-access articles. The results of our study may provide insights into various measures of article impact.

## 1. Introduction

Accurate measurement of the impact of a scientific publication is important for evaluating an individual’s academic performance and for guiding decisions for promotions, tenure, and grant funding [1].

Traditionally, the number of citations an article receives is the most commonly used measure of the impact of an article and may reflect the impact among researchers rather than the interest of a broad audience [2]. A citation is an acknowledgment that a previous article has been referenced in another article. The number of citations is not only an indicator of the impact of an article but also forms the basis of a journal’s impact factor generation [3,4]. Lately, there has been the need to improve the ways in which the outputs of scholarly research are evaluated [5,6]. Another metric for the impact of an article is the download rate. Nowadays, most scientific articles are published not only in print but also in electronic versions of the journal. Therefore, most articles can be downloaded and read online. The download rate of an article may indicate the interest of the readers in a given scientific field, thus reflecting the popularity of paper [7]. With the advent of the Internet and social media, scientific knowledge is now being disseminated to a much larger and more diverse audience. Alternative metrics, also known as altmetrics, are the measure of the real-time online impact of an article and may reflect the interest of a broad audience [8,9,10]. Altmetric.com is one of the main providers of alternative indicators that track where published research is mentioned on various online sources and generate an Altmetric Attention Score (AAS).

Although there have been many studies to identify the most influential articles in a specific biomedical field, to the best of our knowledge, no study has compared the most cited, most downloaded, and most mentioned articles [11,12,13,14,15,16,17,18]. The aim of this study, therefore, was to evaluate and compare the characteristics of the most cited, most downloaded, and most mentioned (specifically, the highest AAS) articles published in general medical journals.

## 2. Methods and Materials

The present study did not involve human subjects and, thus, did not require institutional review board approval.

### 2.1. Selection of Journals and Articles

A search of the Web of Science (Thomson Reuters, New York, NY, USA), Science Edition 2016 was conducted which identified 155 journals listed under the subject category “Medicine, General and Internal”. Of these, 107 journals were excluded from the analysis because they did not display the lists of most downloaded articles (*n* = 8), most cited articles (*n* = 12), or both (*n* = 87) on their websites. Consequently, 48 general medical journals that provided information on both the most cited and most downloaded articles were included in the study. An author (J.H.H.) confirmed that each journal covers a broad spectrum of topics throughout the medical field, based on PubMed’s “Abridged Index Medicus (AIM or ‘Core Clinical’) Journal Titles” and on “Instructions for Authors” in the journal’s website. Authors visited websites of 48 general medical journals, which displayed the list of the most cited and downloaded articles. The total sum of the most cited and downloaded articles from 48 general medical journals were 640 and 662, respectively.

For selection of the most mentioned articles, the AAS provided by Altmetric.com was chosen because Altmetric data are the most comprehensive source covering the vast majority of social media activity associated with scientific papers. The AAS reflects a weighted total of the mentions of the article by the various online platforms. For example, a news story is worth eight points, a tweet is worth one point, and a Facebook post is worth a quarter of a point [9]. For the purpose of comparative analysis, the number of articles with the highest AAS was set as the average of the most cited and most downloaded articles from each journal (after rounding off the average number in each journal). Finally, the 652 articles with the highest AAS published in the 48 journals were identified by an Altmetric Explorer search (Altmetric LLP, London, UK). All searches were conducted on a specific day, 19 May 2018, to avoid changes in lists of articles. No restrictions were applied regarding language, type of study, or scholarly identifiers.

### 2.2. Data Extraction from Articles

For analyzing the characteristics of the most cited, most downloaded, and most mentioned articles, the following information was obtained from each article: (1) medical specialty, (2) publication type (original article, review, case report, guideline/consensus statement, editorial/commentary, systematic review/meta-analysis, or others), (3) country of origin, (4) year of publication, and (5) accessibility (open-access or pay-for-access). Original articles were considered reports that investigated clearly stated objectives or hypotheses and contained original data, precisely articulated methods, and results sections. When an article had multiple authors from different countries, the country of the first author was considered the country of origin. If the first author had affiliations that were in more than one country, the corresponding author’s country of origin was used.

Two reviewers (J.H.H. and K.J.L.) independently reviewed the titles and abstracts and extracted data from the articles. When relevant information was not available in the abstract, the full text of the article was obtained. In cases of disagreement, a third reviewer (D.Y.Y.) was included in the discussion until consensus was achieved.

### 2.3. Statistical Analysis

The characteristics between the most cited, most downloaded, and most mentioned articles were compared using the Chi-Square test. Statistical analyses were performed using IBM SPSS Statistics for Windows (Version 20.0; IBM Corp., Armonk, NY, USA), and a *p*-value of <0.05 was considered statistically significant.

## 3. Results

Among the 640 most cited, the 662 most downloaded, and the 652 most mentioned articles published in the general medical journals, only 16 (2.5%) articles were included in all three lists. Thirty-seven (5.7%) articles were featured in the lists for both the most cited and most downloaded articles, 61 (9.3%) for the most downloaded and most mentioned articles, and 21 (3.3%) for the most mentioned and most cited articles.

Health care was the more frequent specialty among the most cited articles (*p* < 0.0001), and gastroenterology/hepatology was more frequent among the most downloaded articles (*p* < 0.05) (Table 1). Original article was more frequent among the most mentioned articles (*p* < 0.0001), whereas reviews, case reports, and guidelines/consensus statements were more frequent among the most downloaded articles (*p* < 0.0001 for all) (Table 2). The country of origin was most frequently the United States in the most cited, downloaded, and mentioned articles, with no significant difference (34.8% vs. 36.9% vs. 35.1%, respectively, *p* = 0.7). In almost all countries, there were no significant differences between the three groups (Table 3).

The most cited articles were published significantly more frequently in 2010 and before (*p* < 0.0001), the most downloaded articles were published in 2017−2018 (*p* < 0.0001), and the most mentioned articles were published in 2013−2014 and 2015−2016 (*p* < 0.005 for both) (Table 4). Regarding the accessibility, the most mentioned articles were more frequently open-access articles compared to the most downloaded articles (77.5% vs. 70.2%, *p* < 0.05) (Table 4).

## 4. Discussion

This study has several potential limitations. First, the survey was limited to lists of the most cited and most downloaded articles on the journals’ website; in most journals, however, no data are available on the period for citation and download of articles. All metrics for individual articles may fluctuate over time. Second, it was not possible to analyze the correlation between the number of citations, download count, and AAS of articles. Most general medical journals included in this study did not provide the number of times each article was cited and downloaded by readers. Third, only data supplied by Altmetric.com was used for assessing alternative metrics given its wide availability and coverage of the most comprehensive social media data. Other tools that provide article-level alternative metrics, such as Plum Analytics, ImpactStory, and ALM-PLoS, were not evaluated. These altmetric tools use different online sources and algorithms, which may lead to different results [19]. In addition, there are several citation databases such as Web of Science, Scopus (Elsevier, Amsterdam, The Netherlands), and Google Scholar (Google Inc., Mountain View, CA, USA). Thus, it is possible that the citation count from these databases differs. In our study, we sought to use Web of Science, which has been shown to be the most commonly used and robust method for clinical medicine.

The number of citations has been widely used as an accurate measure of the impact of an individual article. However, citation rate has been criticized because older articles have a greater citation potential simply because of the length of time they have been in the public domain. Scientific articles usually are not cited until one or two years after publication, and citations reach a maximum after 3–10 years [20]. Therefore, recently published articles may not have had sufficient time to accumulate citations, despite their scientific originality and impact. In addition, the number of citations can be influenced by many different factors such as obliteration by incorporation (i.e., the phenomenon that information from highly influential articles has been incorporated into common knowledge such that they are not explicitly cited), omission bias (i.e., tendency to not cite competitors or sources contradictory to one’s own results), self-citation, referencing high impact factor core journals, and national or language preferences [21].

Scientific publications are the primary channel for the widespread dissemination of research findings. Therefore, it is likely that the number of downloads of articles provides a good indicator of the papers that are most popular with readers [7]. Several years are needed to observe the initial impact of a new article by means of citation count, but the download rate significantly shortens this lead time as a metric. A potential limitation is that download rate may not represent the true access rate of articles by excluding the access of readers who use the print version of the journal.

Altmetrics provides a real-time measure of the impact of articles on both researchers and the public through social media. Altmetrics, therefore, indicate a wider range of research-related activities than citations and downloads provide [22]. However, altmetrics do not cover the demographics of those mentioning online research material (the ‘’who’’), the reason for and meaning behind online mentions (the ‘’why’’), and the nature and importance of each mention (‘’weight’’). Therefore, the validity of altmetric scoring is hindered because of the possibility of “false papers” receiving undeserved mentions and “self-mentions” by the authors of the paper. Furthermore, the altmetric score can be affected by the presence of the journal’s online content such as dedicated Twitter profiles [23].

Several previous studies have examined the relationship between citation counts and various altmetric measures for scientific papers [24,25,26,27,28]. Most studies have shown very weak correlations between citation rates and altmetric measures [24,26,27]. Among the various indicators, Mendeley readership count was most strongly correlated with citation counts of articles [25,28]. These findings suggest that altmetrics operate independently of traditional metrics (citation and download metrics), and thus can be used as an alternative measure of research impact.

Before the appearance of the altmetrics concept, several studies have investigated the relationship between citations and downloads of publications and found varying results. Early studies have shown that the rank correlation between downloads and citations was high at the journal level but only medium-sized at the article level [29,30]. However, a recent study demonstrated that citation count was strongly correlated with access count (the sum of HTML views and PDF downloads) of articles published in a medical education journal [31]. The likely explanation is that articles are more likely to be downloaded by researchers in the same field for subsequent citation rather than by laypeople.

In this study, the characteristics of the most cited, most downloaded, and most mentioned articles published in general medical journals were compared, and substantial differences were found between these three groups.

In terms of the medical specialty, “health care” specialty was significantly more frequent for the most cited articles, whereas “miscellaneous” was significantly more frequent for the most mentioned articles. Although the comparison of medical specialty should be interpreted with caution because of the numerous kinds of specialties which can lead to a small number of articles in each specialty, these findings suggest that citations are more related in general medical topics compared to other groups. In contrast, it is possible that a wide spread public audience is likely to share medical knowledge across diverse specialties rather than major medical specialties.

The distribution of publication type of articles shows that reviews, case reports, and guidelines/consensus statements were more frequent among the most downloaded articles than other groups. This result suggests that readers of medical journals (e.g., physicians, trainees or medical students) are more interested in articles with an educational purpose. This is consistent with our previous research, which showed that the most downloaded articles published in radiology journals were more frequently reviews, case reports, guidelines/consensus statements, editorials/commentaries, and pictorial essays, compared with the most cited articles [32]. On the other hand, original article was more frequent among the most mentioned articles than other groups. This may be explained by the fact that original articles with the new results and conclusion attract greater attention from both academics and non-academics (the general public) on social media and other online platforms. However, these results are in contrast to findings of a previous study that compared citations indicators and social media counts of all papers published in 2012 and reported that reviews, editorial materials, news items, and letters are more popular on Twitter [33]. The reason for this discrepancy may be due to the difference of study design: our study only included the most mentioned articles published in general medical journals.

Regarding the year of publication, 51.9% (332/640) of the most cited articles were published in 2010 and before, whereas 33.8% (224/662) of the most downloaded articles and 27.3% (178/652) of the most mentioned articles were published in 2017–2018. It takes a long time for citations to accumulate because to receive a citation, the citing articles must be written, edited, peer-reviewed, and finally published. In contrast, media has a function to announce new knowledge, so recent articles tend to be frequently mentioned on social media and, consequently, altmetrics accrue at a much faster rate than for citations [24]. An interesting finding of this study is that the proportion of articles published in 2017–2018 was significantly higher in the most downloaded articles (33.8%, 224/662) than in the most mentioned articles (27.3%, 178/652). Based on this result, the authors postulate that the download rate is the most rapid measure of publication impact compared with not only citation number but also altmetric score.

Several previous studies have reported the advantage of open access in terms of article citations and downloads [7]. Our study verified that more than 70% of the most cited and most downloaded articles were open-access articles. Our analysis also found that the most mentioned articles had the highest proportion of open-access articles (77.5%, 505/652), compared with the most cited and most downloaded articles. A possible explanation for this result is that articles published in open-access journals are widely available for the general population and are of high visibility in various online channels, whereas those published in pay-for-access journals are geared toward professional audiences [34]. In addition, institutional journal subscription access may also affect the dissemination of research within scientific community. Therefore, articles published in prestigious journals with high impact factor should have more citations and downloads.

On the whole, our results indicate that there are important disparities between the most cited, most downloaded, and most mentioned articles. In addition, only 16 (2.5%) articles overlapped across the three groups. Therefore, the number of citations, download rate, and AAS should be considered as tools with different meanings in assessing the impact of scientific publications. Given faster availability and reflection of a broader audience, download rate and altmetric scores represent interesting and relevant indices of article impact, complementary to citation indicators. By combining these metrics, a more holistic assessment of the scientific and social impact of research can be substantiated.

## 5. Conclusions

The most cited were more frequently older, the most downloaded were more frequently recent and educational, and the most mentioned were more frequently original and open-access articles. The results of our study may provide insights into various measures of article impact.

## Figures and Tables

**Table 1 healthcare-08-00492-t001:** Medical specialties of the most cited, most downloaded, and most mentioned articles published in general medical journals.

Medical Specialty	Most Cited Articles (*n* = 640)	Most Downloaded Articles (*n* = 662)	Most Mentioned Articles (*n* = 652)	*p* Value
Health care	258 (40.3) ^‡^	185 (27.9)	163 (25.0)	<0.0001
Hematology/oncology	47 (7.3)	40 (6.0)	34 (5.1)	0.3
Endocrinology	46 (7.2)	40 (6.0)	44 (6.7)	0.7
Neuroscience	38 (5.9)	71 (10.7) ^†^	77 (11.8) ^†^	0.0007
Infection	38 (5.9)	35 (5.3)	36 (5.5)	0.9
Cardiology	29 (4.5)	49 (7.4) ^†^	32 (4.9)	<0.05
Pulmonology	17 (2.7)	14 (2.1)	14 (2.1)	0.8
Pain medicine	12 (1.9)	25 (3.8) ^†^	28 (4.3) ^†^	0.04
Gastroenterology/hepatology	14 (2.1)	29 (4.4) *	15 (2.3)	0.03
Miscellaneous	141 (22.0)	174 (26.3)	209 (32.1) ^§^	0.0002

Note: Numbers in parentheses are percentages. * Significantly higher than the most cited and most mentioned articles. ^†^ Significantly higher than the most cited articles. ^‡^ Significantly higher than the most downloaded and most mentioned articles. ^§^ Significantly higher than the most cited and most downloaded articles.

**Table 2 healthcare-08-00492-t002:** Publication type of the most cited, most downloaded, and most mentioned articles published in general medical journals.

Publication Type	Most Cited Articles (*n* = 640)	Most Downloaded Articles (*n* = 662)	Most Mentioned Articles (*n* = 652)	*p* Value
Original article	344 (53.8) ^†^	232 (35.0)	449 (68.9) ^‡^	<0.0001
Review	191 (29.8) ^§^	249 (37.6) ^ǁ^	122 (18.7)	<0.0001
Case report	7 (1.1)	56 (8.5) ^ǁ^	27 (4.1) ^¶^	<0.0001
Guideline/consensus statement	9 (1.4)	29 (4.4) ^ǁ^	3 (0.5)	<0.0001
Editorial/commentary	47 (7.3) ^§^	41 (6.2)	27 (4.1)	<0.05
Systematic review/meta-analysis	36 (5.6) ^§^	31 (4.7)	18 (2.8)	0.04
Others *	6 (0.9)	24 (3.6) ^ǁ^	6 (0.9)	0.0002

Note: Numbers in parentheses are percentages. * Include technical note, pictorial essay, letter, news, abstract, correction, and book review. ^†^ Significantly higher than the most downloaded articles. ^‡^ Significantly higher than the most cited and most downloaded articles. ^§^ Significantly higher than the most mentioned articles. ^ǁ^ Significantly higher than the most cited and most mentioned articles. ^¶^ Significantly higher than the most cited articles.

**Table 3 healthcare-08-00492-t003:** Countries of origin of the most cited, most downloaded, and most mentioned articles published in general medical journals.

Country/Region	Most Cited Articles (*n* = 640)	Most Downloaded Articles (*n* = 662)	Most Mentioned Articles (*n* = 652)	*p* Value
United States	223 (34.8)	244 (36.9)	229 (35.1)	0.7
United Kingdom *	104 (16.3)	104 (15.7)	116 (17.8)	0.6
Canada	61 (9.5)	64 (9.7)	59 (9.0)	0.9
China ^†^	39 (6.1)	36 (5.4)	36 (5.5)	0.9
Turkey	17 (2.7)	29 (4.4)	28 (4.3)	0.2
Korea	27 (4.2)	22 (3.3)	24 (3.7)	0.7
Nigeria	24 (3.8)	21 (3.2)	19 (2.9)	0.7
The Netherlands	17 (2.7)	12 (1.8)	14 (2.1)	0.6
India	10 (1.6)	13 (2.0)	16 (2.5)	0.5
Belgium	13 (2.0)	12 (1.8)	10 (1.5)	0.8
Taiwan	11 (1.7)	9 (1.4)	15 (2.3)	0.4
Sweden	8 (1.3)	10 (1.5)	11 (1.7)	0.8
Australia	14 (2.2) ^‡^	2 (0.3)	6 (0.9)	0.005
Others	72 (11.3)	84 (12.7)	69 (10.6)	0.5

Note: For the purpose of our research, country of origin was defined by the address provided for the first author. If the first author was affiliated with more than one institution or group name, the corresponding author’s affiliation was used for the origin of the article. Numbers in parentheses are percentages. * Includes articles originating from England, Scotland, Wales, and Northern Ireland. ^†^ Includes articles originating from Hong Kong. ^‡^ Significantly higher compared to the most downloaded articles.

**Table 4 healthcare-08-00492-t004:** Publication year and accessibility of the most cited, most downloaded, and most mentioned articles published in general medical journals.

Publication Information	Most Cited Articles (*n* = 640)	Most Downloaded Articles (n = 662)	Most Mentioned Articles (n = 652)	*p* Value
Publication Year				
−2010	332 (51.9) *	154 (23.3) ^†^	113 (17.3)	<0.0001
2011−2012	88 (13.8)	76 (11.5)	69 (10.6)	0.2
2013−2014	79 (12.3)	76 (11.5)	116 (17.8) ^‡^	0.002
2015−2016	121 (18.9)	132 (19.9)	176 (27.0) ^‡^	0.0006
2017−2018	20 (3.1)	224 (33.8) ^§^	178 (27.3) ^ǁ^	<0.0001
Accessibility				0.01
Open access	473 (73.9)	465 (70.2)	505 (77.5) ^¶^	
Pay-for-access	167 (26.1)	197 (29.8)	147 (22.5)	

Note: Numbers in parentheses are percentages. * Significantly higher than the most downloaded and most mentioned articles. ^†^ Significantly higher than the most mentioned articles. ^‡^ Significantly higher than the most cited and most downloaded articles. ^§^ Significantly higher than the most cited and most mentioned articles. ^ǁ^ Significantly higher than the most cited articles. ^¶^ Significantly higher than the most downloaded articles.

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
