# Peer review of "Characteristics of the Most Cited, Most Downloaded, and Most Mentioned Articles in General Medical Journals: A Comparative Bibliometric Analysis"

_healthcare, 2020, doi:10.3390/healthcare8040492_

Round 1
Reviewer 1 Report
Overall, this is a great article. The content and conclusions appear to be solid, and are especially reinforced with the inclusion/exclusion methodology mentioned in Lines 92-95. Additionally, the content and organization of the Tables is strength of this article.
However, there are a few suggestions for improvement, or to be considered for future bibliometric publications:
Major Suggestions
- The "Key Points" and "Introduction" sections can be combined into one section, especially since they are both trying to convey basically the same information.
- The "Results" section needs to be broken into two different sections - "Results" and "Conclusion".
- Even though you appropriately self-cited (Lines 202-205 and Citation 28) based on the point of the sentence, you need to identify that you in fact did self-cite. Maybe start the line similar to this: "In our previous research, we ...". Otherwise, this could be considered an inappropriate self-citation.
Minor Suggestions/Grammatic Changes
- Line 32: consider adding "... in medical journals." to the end of the sentence for additional clarification.
- Line 33: "sited" needs to be changed to "cited".
- Line 63-64: For future consideration, please also refer to PubMed's "Abridged Index Medicus (AIM or 'Core Clinical') Journal Titles" (URL: https://www.nlm.nih.gov/bsd/aim.html) as another source for identifying core biomedical journal titles. The ISI list may not be comprehensive enough, based only on the journal impact factors.
- Lines 82-95. Section "3.2 Data extraction form articles": A question for clarification - did any of the authors note if any of the articles included happened to be retracted publications?. i tois possible to elevate altmetrics impact if an article was retracted. This could be pushed up by by increasing social media mentions and viewership, just based on the news that article has been retracted. If this data is not readily available, please keep this in consideration for future bibliometric analyses.
- In lines 245-246, you mention why additional altmetric tools were not used, but in the paragraph for Lines 175-180, you mention Mendeley readership count, but do not address why citation counts from other databases, such as Scopus (https://www.scopus.com/), Google Scholar, and to a lesser degree, PubMed, were not included. Some clarification around the database exclusion would be very helpful. You do not have to go back and reanalyze your data regarding these citation resources though.
- Lines 227-229. The open access explanation is a valid one, but it overlooks institutional journal subscription access, through the library, as a factor. If you could also address this factor, you will have a complete argument.
- In the "Discussion" section, you use percentages without any accompanying numerical data. Inclusion of numerical data along with the percentages will make this section easier to read.
I recommend this article for publication, as long as all 3 of my "Major Suggestions" are effectively addressed. Additionally, if at least a few of my "Minor Suggestions" are addressed, the article will only be stronger.
Author Response
Reviewer#1
Overall, this is a great article. The content and conclusions appear to be solid, and are especially reinforced with the inclusion/exclusion methodology mentioned in Lines 92-95. Additionally, the content and organization of the Tables is strength of this article.
However, there are a few suggestions for improvement, or to be considered for future bibliometric publications:
Major Suggestions
- The "Key Points" and "Introduction" sections can be combined into one section, especially since they are both trying to convey basically the same information.
Reply) According to “Instructions for Authors” of Healthcare journal, research manuscript sections should comprise the following sections: Introduction, Materials and Methods, Results, Discussion, Conclusions (optional). The <Key points> section in the manuscript is obvious mistake of authors. We are terribly sorry about that. We have removed the <Key points> section.
- The "Results" section needs to be broken into two different sections - "Results" and "Conclusion".
Reply) As the reviewer suggested, we have added the <Conclusion> section, to read as follows:
[Revised manuscript] Line 281-287
The most cited were more frequently older, the most downloaded were more frequently recent and educational, and the most mentioned were more frequently original and open access articles. The results of our study may provide insights into various measures of article impact.
- Even though you appropriately self-cited (Lines 202-205 and Citation 28) based on the point of the sentence, you need to identify that you in fact did self-cite. Maybe start the line similar to this: "In our previous research, we ...". Otherwise, this could be considered an inappropriate self-citation.
Reply) According to the recommendation, we have revised the sentence about self-citation.
[Revised manuscript] Lines 228-231
This is consistent with our previous research, which showed that the most downloaded articles published in radiology journals were more frequently reviews, case reports, guidelines/consensus statements, editorials/commentaries, and pictorial essays compared with the most cited articles 32.
Minor Suggestions/Grammatic Changes
- Line 32: consider adding "... in medical journals." to the end of the sentence for additional clarification.
Reply) Please see response to major suggestion 1 above.
- Line 33: "sited" needs to be changed to "cited".
Reply) Please see response to major suggestion 1 above.
- Line 63-64: For future consideration, please also refer to PubMed's "Abridged Index Medicus (AIM or 'Core Clinical') Journal Titles" (URL: https://www.nlm.nih.gov/bsd/aim.html) as another source for identifying core biomedical journal titles. The ISI list may not be comprehensive enough, based only on the journal impact factors.
Reply) In agreement of the reviewer, we have added the following sentence in the <Materials and Method> section:
[Revised manuscript] Line 71-74
...... included in the study. An authors (J.H.H.) confirmed that each journal covers a broad spectrum of topics throughout the medical field, based on PubMed's "Abridged Index Medicus (AIM or 'Core Clinical') Journal Titles" (URL: https://www.nlm.nih.gov/bsd/aim.html) and on “Instructions for authors” in the journal's website. A total of the 640......
- Lines 82-95. Section "3.2 Data extraction form articles": A question for clarification - did any of the authors note if any of the articles included happened to be retracted publications?. i tois possible to elevate altmetrics impact if an article was retracted. This could be pushed up by by increasing social media mentions and viewership, just based on the news that article has been retracted. If this data is not readily available, please keep this in consideration for future bibliometric analyses.
Reply) For data extraction, we reviewed both titles and abstracts of all articles. However, there was no article marked as “retracted article”. As far as we know, retracted publications are clearly watermarked (or other information that may be embedded) to indicate that the article was retracted. Thus, we estimated that our data did not include any retracted article.
- In lines 245-246, you mention why additional altmetric tools were not used, but in the paragraph for Lines 175-180, you mention Mendeley readership count, but do not address why citation counts from other databases, such as Scopus (https://www.scopus.com/), Google Scholar, and to a lesser degree, PubMed, were not included. Some clarification around the database exclusion would be very helpful. You do not have to go back and reanalyze your data regarding these citation resources though.
Reply) Altmetric Attention Scores are composite scores that include data from many social media platforms (e.g. Twitter, Facebook, Wikipedia, Mendeley, etc.). Therefore, Mendeley readership count is a component for calculating Altmetric Attention Score.
In agreement of the reviewer, we have added the following sentences in the limitation part of the <Discussion> section:
[Revised manuscript] Line 170-174
In addition, there are several citation databases such as Web of Science, Scopus (Elsevier, Amsterdam, The Netherlands), and Google Scholar (Google Inc., Mountain View, CA). Thus, it is possible that the citation count from these databases differs. In our study, we sought to use Web of Science, which has been shown to be the most commonly used and robust method for clinical medicine.
- Lines 227-229. The open access explanation is a valid one, but it overlooks institutional journal subscription access, through the library, as a factor. If you could also address this factor, you will have a complete argument.
Reply) In agreement of the reviewer, we have added the following sentence in the <Discussion section>.
[Revised manuscript] Line 255-260
A possible explanation for this result is that articles published in open access journals are widely available for the general population and are of high visibility in various online channels, whereas those published in pay-for-access journals are geared toward professional audiences. In addition, institutional journal subscription access may also affect the dissemination of research within scientific community. Therefore, articles published in prestigious journals with high impact factor should have more citations and downloads.
- In the "Discussion" section, you use percentages without any accompanying numerical data. Inclusion of numerical data along with the percentages will make this section easier to read.
Reply) As the reviewer recommended, we have added numerical data along with percentages in the <Discussion> section.
[Revised manuscript] Line 240-242
Regarding the year of publication, 51.9% (332/640) of the most cited articles were published in 2010 and before, whereas 33.8% (224/662) of the most downloaded articles and 27.3% (178/652) of the most mentioned articles were published in 2017–2018.
[Revised manuscript] Line 246-248
An interesting finding of this study is that the proportion of articles published in 2017–2018 was significantly higher in the most downloaded articles (33.8%, 224/662) than in the most mentioned articles (27.3%, 178/652).
[Revised manuscript] Line 253-255
Our analysis also found that the most mentioned articles had the highest proportion of open access articles (77.5%, 505/652) compared with the most cited and most downloaded articles.
I recommend this article for publication, as long as all 3 of my "Major Suggestions" are effectively addressed. Additionally, if at least a few of my "Minor Suggestions" are addressed, the article will only be stronger.
Reviewer 2 Report
The manuscript is well written and easy to understand. Below are some specific comments.
The conclusion in abstracts is unclear and needs to be rewritten.
Please clarify how to calculate the sample size (most cited= 640, most downloaded= 662, and most mentioned=652 articles)?
In the method section, clarify why the search is limited to 2016.
If compared to quality assessments of three groups of articles, please mention?
The references list needs to be updated. There are old references that can be replaced with more recent ones.
Author Response
Reviewer#2
The manuscript is well written and easy to understand. Below are some specific comments.
- The conclusion in abstracts is unclear and needs to be rewritten.
Reply) As the reviewer suggested, we have revised the conclusion of <Abstract> section, to read as follows:
[Revised manuscript] Line 23-26
The most cited were more frequently older, the most downloaded were more frequently recent and educational, and the most mentioned were more frequently original and open access articles. The results of our study may provide insights into various measures of article impact.
- Please clarify how to calculate the sample size (most cited= 640, most downloaded= 662, and most mentioned=652 articles)?
Reply) We have added more sentences to explain how the sample size (most cited= 640, most downloaded= 662, and most mentioned=652 articles) were determined in this study.
[Revised manuscript] Line 75-77
Authors visited wed-sites of 48 general medical journals, which displayed the list of the most cited and downloaded articles. The total sum of the most cited and downloaded articles from 48 general medical journals were 640 and 662, respectively.
[Revised manuscript] Line 83-88
For the purpose of comparative analysis, the number of articles with the highest AAS was set as the average of the most cited and most downloaded articles from each journal (after rounding off the average number in each journal). Finally, the 652 articles with the highest AAS published in the 48 journals were identified by an Altmetric Explorer search (https://www.altmetric.com/explorer, Altmetric LLP, London, UK).
- In the method section, clarify why the search is limited to 2016.
Reply) We started this study in May 2018. In that time, Web of Science, Science Edition 2017 and 2018 were not available. Therefore, we used Science Edition 2016 to identify general medical journals listed under the subject category “Medicine, General and Internal”. We had no time restriction in the selection of the most cited, most downloaded, and most mentioned articles.
- If compared to quality assessments of three groups of articles, please mention?
Reply) Although there have been various attempts to assess the quality of the article, there is no standardized or objective way to assess it. Thus, we did not assess the quality of the article.
- The references list needs to be updated. There are old references that can be replaced with more recent ones.
Reply) As the reviewer recommend, we updated several old references into more recent ones. In our references, some of the milestone publications are old articles but have contributed greatly to the field of scientometrics.
[Revised manuscript] Line 302-304
- Garfield, E., 100 citation classics from the Journal of the American Medical Association. JAMA 1987, 257 (1), 52-9. à Zhang, W. J.; Li, Y. F.; Zhang, J. L.; Xu, M.; Yan, R. L.; Jiang, H., Classic citations in main plastic and reconstructive surgery journals. Ann Plast Surg 2013, 71 (1), 103-8
[Revised manuscript] Line 323-325
- Paladugu, R.; Schein, M.; Gardezi, S.; Wise, L., One hundred citation classics in general surgical journals. World J Surg 2002, 26 (9), 1099-105. à Fardi, A.; Kodonas, K.; Gogos, C.; Economides, N., Top-cited articles in endodontic journals. J Endod 2011, 37 (9), 1183-90.
Reviewer 3 Report
This is an interesting paper about academic impact. Howevern I would suggest some improvements before final acceptation.
Introduction
Two aspects should be included: the political and academic context of open science and new forms of scientific evaluation (cf. Jason Priem's Altmetrics Manifesto 2010 and the DORA declaration); and the principal component analysis of different impact measures by Bollen et al. 2009 (https://doi.org/10.1371/journal.pone.0006022) which provides evidence on the difference between citation and usage metrics.
Methodology
The sampling method remains unclear: why 640/662/652 articles? The provided information does not allow a replication. Also, information about the three samples is missing (eg, journal titles with articles). How were the papers' domains indexed (referential)? How was the accessibility assessed - Unpaywall? When? By the way: is the dataset available (data repository)?
Results
The description of the results is very (too) short and superficial. Table 1 - the order of the categories makes no sens, there should be another ranking. Table 4 - these results are important for the discussion; yet the basic interpretation seems a little bit simple - that older papers are more cited than recent papers (one can't expect citations of 2018 papers in a 2018 study), and that recent papers are more mentioned in social networks than older... When the older papers (<2010) and the recent (2017-2018) papers are skipped, the statistical test reveals (p=0.05) that the main difference is only with the older papers (2011-2012), between citations (+) and mentions (-).
Discussion
The section should be clearly structured in different topics, starting with the limitations. The results should be better compared with other studies, from medical fields and from other disciplines. Another question: does the relationship between the three metrics change with the years? What can be said about this evolution? Finally, the term of impact should be discussed, also in the context of open science.
Conclusion
Is missing and should be added, with main results, implications and perspectives for further research.
Author Response
Reviewer#3
This is an interesting paper about academic impact. Howevern I would suggest some improvements before final acceptation.
- Introduction: Two aspects should be included: the political and academic context of open science and new forms of scientific evaluation (cf. Jason Priem's Altmetrics Manifesto 2010 and the DORA declaration); and the principal component analysis of different impact measures by Bollen et al. 2009 (https://doi.org/10.1371/journal.pone.0006022) which provides evidence on the difference between citation and usage metrics.
Reply) As the reviewer suggested, we have added a sentence (and three references) in the <Introduction> section, to read as follows:
[Revised manuscript] Line 46-47
Lately, there has been the need to improve the ways in which the outputs of scholarly research are evaluated. 5, 6
- DORA – San Francisco Declaration on Research Assessment (DORA). https://sfdora.org/read/ (accessed on 09 November 2020).
- Bollen, J.; Van de Sompel, H.; Hagberg, A.; Chute, R., A principal component analysis of 39 scientific impact measures. PLoS One 2009, 4 (6), e6022.
[ Revised manuscript] Line 52-54
The alternative metrics, also known as altmetrics, are the measure of the real-time online impact of the article and may reflect the interest of a broad audience8-10.
- Altmetrics: A manifesto. http://altmetrics.org/manifesto (accessed on 09 November 2020).
- Methodology: The sampling method remains unclear: why 640/662/652 articles? The provided information does not allow a replication. Also, information about the three samples is missing (eg, journal titles with articles). How were the papers' domains indexed (referential)? How was the accessibility assessed - Unpaywall? When? By the way: is the dataset available (data repository)?
Reply) We have added more sentences to explain how the sample size (most cited= 640, most downloaded= 662, and most mentioned=652 articles) were determined in this study, to read as follows:
[Revised manuscript] Line 75-77
Authors visited wed-sites of 48 general medical journals, which displayed the list of the most cited and downloaded articles. The total sum of the most cited and downloaded articles from 48 general medical journals were 640 and 662, respectively.
[Revised manuscript] Line 83-88
For the purpose of comparative analysis, the number of articles with the highest AAS was set as the average of the most cited and most downloaded articles from each journal (after rounding off the average number in each journal). Finally, the 652 articles with the highest AAS published in the 48 journals were identified by an Altmetric Explorer search (https://www.altmetric.com/explorer, Altmetric LLP, London, UK).
Reply) If needed, we can provide all of journal titles and articles as supplemental tables.
- Results: The description of the results is very (too) short and superficial. Table 1 - the order of the categories makes no sens, there should be another ranking. Table 4 - these results are important for the discussion; yet the basic interpretation seems a little bit simple - that older papers are more cited than recent papers (one can't expect citations of 2018 papers in a 2018 study), and that recent papers are more mentioned in social networks than older... When the older papers (<2010) and the recent (2017-2018) papers are skipped, the statistical test reveals (p=0.05) that the main difference is only with the older papers (2011-2012), between citations (+) and mentions (-).
Reply) As the reviewer pointed out, we have rearranged of the categories in the descending order of numbers of articles.
[Revised manuscript] Table 1, Line 132-137
Reply) Although there was no statistical significance when compared three metrics in the periods of 2011-2012, 2013-2014, and 2015-2016, we believe that overall differences in publication year were determined among three groups using statistical methods.
- Discussion: The section should be clearly structured in different topics, starting with the limitations. The results should be better compared with other studies, from medical fields and from other disciplines.
Reply) As suggested, we have revised the discussion section, starting with the limitations.
[Revised manuscript] Line 160-174
Reply) To the best of our knowledge, there has been only one study comparing the characteristics of the most-cited and most-downloaded articles in the field of radiology (reference number: 32) and no study comparing the characteristics of the most cited, most downloaded, and most mentioned (the highest Altmetric Attention Score) articles in any scientific discipline. Several previous studies have examined the relationship between citation counts and various altmetric measures for scientific papers and shown very weak correlations between citation rates and altmetric measures.
- Another question: does the relationship between the three metrics change with the years? What can be said about this evolution? Finally, the term of impact should be discussed, also in the context of open science.
Reply) All searches in our study were conducted on a specific day, May 19, 2018. Thus, we have no idea whether the relationship between the three metrics changes with the years. It can be speculated that the widespread use of the Internet and online resources provide a vast array of means of interaction, which may increase the importance of altmetric measure of the scientific publications.
Although we did not evaluate the term of impact of scientific articles in our study, several previous studies reported that an article usually reached its citation peak 2-5 years after publication and thereafter showed a slowly decreasing pattern of citation. It may be difficult to assess the time-trend of downloading and altmetric score, because of insufficient information. Further study is warranted on this topic.
- Conclusion is missing and should be added, with main results, implications and perspectives for further research.
Reply) As the reviewer suggested, we have added <Conclusion> section and revised the conclusion of the paper, to read as follows:
[Revised manuscript] Line 281-287
The most cited were more frequently older, the most downloaded were more frequently recent and educational, and the most mentioned were more frequently original and open access articles. The results of our study may provide insights into various measures of article impact.